# Physiological Reactions in the Therapist and Turn-Taking during Online Psychotherapy with Children and Adolescents with Autism Spectrum Disorder

**DOI:** 10.3390/brainsci11050586

**Published:** 2021-04-30

**Authors:** Laura López-Florit, Esteban García-Cuesta, Luis Gracia-Expósito, German García-García, Giuseppe Iandolo

**Affiliations:** 1Department of Psychology, School of Biomedical Sciences, European University of Madrid, Calle Tajo S/N., Villaviciosa de Odón, 28670 Madrid, Spain; laura.lopezflorit@psisemadrid.com; 2Observation and Functional Diagnosis Division, PSISE Clinical and Developmental Psychological Service, Calle Albendiego 7, 28029 Madrid, Spain; german.garcia@psisemadrid.com; 3Department of Science, Computing, and Technology, School of Architecture, Engineering and Design, European University of Madrid, Calle Tajo S/N., Villaviciosa de Odón, 28670 Madrid, Spain; luis.gracia@universidadeuropea.es

**Keywords:** autism spectrum disorder, heart rate variability, turn-taking, online psychotherapy, synchrony

## Abstract

This study aims to analyze the relationship between the sociocognitive skills of a group of children and adolescents with autism spectrum disorder (ASD) at verbal level 1, the variability of the therapist’s heart rate (HRV), and the conversational turn-taking during online psychotherapy sessions. Initially, we assessed the intelligence, narrative, and behavioral characteristics of the participants. We videotaped the online sessions and recorded the therapist’s HRV via a smart wireless sensor. Finally, we analyzed the video sessions using an observation system and the therapist’s HRV using the Poincaré technique. The results show that the patients’ communicative intention was related to their narrative, intellectual and social competencies. Furthermore, the turn-taking between the therapist and the participant was associated with the patient’s emotional and behavioral difficulties. On the other side, the therapist’s heart rate variability (HRV) was related to the synchrony between the therapist and the participant with more significant stress on the therapist, when he shared and expanded the conversation with the patient, and when the patient broadened and shared the conversation with the therapist.

## 1. Introduction

The therapist–patient relationship is a specific intersubjective interaction environment, determined by a high level of confidentiality and emotional involvement [1,2]. It encompasses and respects the same rules and characteristics that appear in all human interactions, where both agents are active parts of the turn-taking process [2].

From this perspective, the synchrony between therapist and young patients can be defined as “responsiveness” [3], “reciprocity” [4], “rhythmicity”, “mutuality” [5,6], “turn-taking”, and “shared affect” [7].

The dyadic interaction between any caregiver and child implies synchrony, a continuous, dynamic, reciprocal behavioral and emotional adaptation during shared activities [1,2,8,9]. The quality of the interaction with the caregiver is crucial for children’s optimal social development [10,11] because it favors emotional and cognitive progression [7,12,13].

Consequently, reciprocal behavioral adaptation and turn-taking are interactive components required in the caregiver–child relationship and the therapeutic one, especially with children and adolescents with social impairments such as autism spectrum disorder. Moreover, turn-taking includes cognitive, behavioral, emotional, and physiological components [14,15,16].

Therefore, quantifying the turn-taking and physiological activity during psychotherapy allows one to understand how the therapist recognizes, regulates, and reorganizes cognitive activity for reciprocal emotional attunement [17], adapting and maximizing the psychotherapeutic intervention [16].

Intersubjectivity, social interaction and turn-taking allow the development of affective bonds, attachment [18,19] and therapeutic alliance [20]. Affective processes are inseparable from cognitive ones, interconnected with communication, and are essential aspects in children and adolescents’ intervention with ASD [21,22,23,24]. The theory of social interaction postulates that the child’s behavior and the parents’ behavior are interdependent [25,26]. Interactions that encourage reciprocity and balanced turn-taking are considered the most conducive to develop children’s communication. Children who show low communication development display frustration and a variety of disruptive behaviors. These behaviors may be due, at least partially, to a break or imbalance in the shift between parents and children, a form of social reciprocity [27]. Parental stress significantly reduces mother–child brain synchrony in the left medial prefrontal cortex when dyads are engaged in a typical joint attention task. Given the importance of the synchrony as a critical co-regulatory process in emotional development, this mechanism may potentially underlie the strong associations between parental stress and children’s externalizing behaviors in later development [7,28].

One of the primary intervention approaches with dysregulation and disruptive behaviors in children and adolescents with ASD focuses on communication [29], on the pragmatic aspects of language [30] and on the teaching of planning skills [31]. According to [32], in ASD, it is essential to work on executive functions and planning, to reduce disruptive, obsessive, and stereotyped behaviors.

From this perspective, ecological and mainstreaming psychotherapy promotes language adaptation and social skills, always considering interlocutors [33,34,35,36,37]. When the therapist leans for symmetry in the relationship with the patient, they yield greater communicative functionality, indispensable for children and adolescents with ASD [38].

From a novel perspective, online psychotherapy offers a flexible patient–therapist setting for synchronic conversations, including the child and adolescent’s daily environment, especially with people with verbal-level ASD [39]. The effect of online disinhibition is a key property of this type of communication, which can favor reducing the prejudice of sharing experiences [40]. Additionally, it offers the possibility of a spontaneous conversation and synchronized communication [41].

Autism spectrum disorders (ASD) are neurodevelopmental disorders with different quantitative and qualitative impairments in communication, interaction, and restricted behaviors/interests [34,42,43,44]. Children and adolescents’ with an ASD neuropsychological profile get through a heterogeneous group of sociocognitive and behavioral clinical impairments [45]. Between 60 and 90% of children with ASD show symptoms related to some difficulty or disorder in sensory modulation [46,47,48,49]. Sensory modulation impairments along with a weak central coherence bias [50], executive functioning deficits [32,51,52,53], impairments in intersubjectivity [18,22] and “theory of mind” [54,55,56,57,58,59,60,61,62] make young people with ASD show different physiological and behavioral responses to sensory and social stimulation than typically developing children [48,63,64].

This sensory particularity and perceptual weak central coherence bias in ASD can generate high levels of anxiety and, consequently, an increase in the manifestation of obsessive and compulsive behaviors [65]. Moreover, they can show two kinds of extreme behavioral response: sensation-seeking or sensory avoidance [66,67,68,69]. Due to these circumstances, young people with ASD can present high levels of stress and frustration with external experiences that can lead to anxiety, depression, or obsessive-type disorders. These factors can lead to various behavioral reactions: disconnection, flight, distraction, stereotypes, disruption, or emotional lability. These alterations can affect the quality of life and learning of children and adolescents with ASD.

Ultimately, understanding the relationship between self-esteem, personal resources, emotional and behavioral difficulties (according to self-report, family, and school), narrative, and social skills of young people with ASD, the turn-taking and the therapist’s physiological variability during discursive dialogic psychotherapy, can be useful to explore the communication, representation, and social adaptation in ASD.

Moreover, given the importance of turn-taking and patient-therapist synchrony, it is crucial to analyze the therapist’s reactions during psychotherapy to make explicit and optimize the interventions with young people with verbal-level ASD.

In this way, the psychotherapy can be calibrated to improve self-awareness and behavioral–emotional self-regulation, optimizing, and promoting more flexible and balanced representations [70].

## 2. Method

### 2.1. Design

The study’s design is cross-sectional, and explores behavioral and emotional characteristics of a group of children and adolescents with ASD, the turn-taking, and the therapist’s heart rate variability (HRV) during 26 online discursive dialogic therapy sessions with 16 Spanish children and adolescents with verbal-level ASD. Several participants’ characteristics were explored, including age, narrative, social skills, and emotional and behavioral difficulties (according to self-report, family, and school). Moreover, we analyzed the turn-taking between the therapist and patient during the online session through sequential observational analysis of video recordings, with minimum intervals of one second. Finally, we defined two different conditions based on the therapist’s electrocardiogram (ECG) signal: condition A (low heart rate variability—HRV), defined by a low sympathetic activity (low-stress), and condition B (high HRV), defined by a high sympathetic activity (high-stress).

Three hypotheses were explored:

**Hypothesis 1** **(H1).**
*The communicative intention during the turn-taking of children/teenagers with ASD and therapist can be related to participants’ intelligence, age, narrative, and social skills.*


**Hypothesis 2** **(H2).**
*The turn-taking between the therapist and the young patients with ASD can be related to the participant’s emotional and behavioral difficulties, according to self-report, family, and school.*


**Hypothesis 3** **(H3).**
*During the psychotherapeutic session, the therapist’s heart rate variability (HRV) can be related to the patient’s turn-taking synchrony, with more significant stress on the therapist throughout shared and broadened activities.*


### 2.2. Participants

The study involved 16 Spanish boys between 6 and 18 years (mean 12,5; SD 299) with verbal-level autism spectrum disorder (ASD), level 1. The participants were selected from a Spanish psychological center with verbal skills to perform a narrative task (Table 1). All the participants received a previous autism spectrum disorder diagnosis based on the DSM-5 [42] through the ADOS or ADOS-2 [71]. All the participants were males; the boy–girl autism gender ratio is usually four to one [72], which is over-represented among high-functioning cases [73,74]. For the study, we also considered the chronological age and the equivalent age based on IQ assessment (mean 12,5; SD 3,29) because some participants were language delayed; hence, equivalent age was a more appropriate index than standard scores [75,76,77]. We analyzed a total of 26 therapeutic sessions, with 16 children and adolescents with ASD with the same therapist.

### 2.3. Instruments

A trained researcher assessed participants’ intelligence, narrative, and behavioral characteristics, in person, using the Reynolds RIAS intelligence test [78], the SENA Children and Adolescents Assessment System [79], and the Bears Family Projective Test [70,80,81,82,83,84].

Each online psychotherapy session (using Skype Video Call; skype.com) was video-recorded (using OBS Studio; obsproject.com) and analyzed by two independent researchers, blinded to the study hypothesis, using a turn-taking sequential observation system [85,86].

The therapist’s heart rate activity was recorded through a smart wireless sensor (Comftech CozyBaby ECG sensor; comftech.com), a smartphone Xiaomi Redmi Note 3, a signal acquisition software (physiolitix.fbk.eu) and an ECG signal processing system [87].

The RIAS intelligence test [79] is an individually applied test, from the age of three to adulthood, that provides a verbal intelligence index (IV), a non-verbal intelligence index (INV), a verbal memory index (IM), and a general IQ (IG).

The child and adolescent evaluation system SENA [79] is a multi-informant questionnaire focused on collecting behavioral, relational, and functional information from three to eighteen years old in different contexts (self-report, family, and school).

The Bears Family Projective Test [70,80,81,82,84,88] is a thematic creative narrative production test with a standard administration and scoring system that allows the emergence and evaluation of storytelling from the age of three to adolescence. The test involves providing the child (play format) or the adolescent (photographic format) with a set of small dolls and dramatic material (or a photography for adolescence) from a family of anthropomorphic bears, for 10 min, to then tell a story in a time limit of 5 min. The test is video-recorded, the story told is transcribed verbatim and analyzed according to the Integrated System of Analysis of the Bears Family [83]. The test provides two report areas: the story form and the story content. The story form assesses the number of propositions, episodes, narrative cohesion, and narrative structure indexes. The story content assesses the number of problematic events with and without a solution, the characters’ use and location, the number and balance of positive and negative relationships between characters, and the number and the balance of adaptive and maladaptive behaviors of the characters of the story.

The turn-taking sequential observation system adapts the caregiver-child turn-taking observation system [85,86]. It is a sequential quantitative observation system that considers the frequencies and duration of each behavior observed, applied to video-recorded sessions by a trained observer, considering minimum sequences of dyadic interactions of one second. The system requires dividing the video-recorded session into three parts: start, middle, and end. In each moment, the observer simultaneously categorizes the therapist’s and the patient’s behavior. Regarding the therapist’s behaviors, the observer can choose between the following ten categories: 1—therapist proposes (T*), 2—therapist holds (TS), 3—therapist does not hold (TNS), 4—co-oriented therapist (TMC), 5—therapist expands (TA), 6—Therapist directs attention (TDAt), 7—Therapist directs action (TDAc), 8—therapist limits (TL), 9—therapist interrupts (TI), and 10—therapist ends (TT). Regarding the patient’s behaviors, the observer can choose between the following eight categories: 1—patient proposes (N*), 2—patient accepts (NA), 3—patient does not accept (NNA), 4—patient shares (NC), 5—patient does not share and proposes (NNcP), 6—patient does not share and continues with his/her game or topic (NNcC), 7—patient Expands (NAm), and 8—patient ends (NT).

The Comftech CozyBaby ECG sensor (comftech.com) is a textile type sensor that allows continuous and non-invasive heart rate variability (HRV) monitoring, reflecting the therapist’s emotional stress level. It measures the ECG signal in mV, through two electrodes (RA—right Arm, and LA—left Arm) that captures 128 samples per second (128 Hz), representing the first ECG lead. This measurement provides the voltage between the left arm electrode and the right arm electros Lead I = LA − RA. The ECG sensor requires both, a signal acquisition and a signal processes. The signal acquisition process pairs the ECG sensor with a smartphone via Bluetooth connection and runs the PhysioREC application (physiolitix.fbk.eu). PhysioREC is a smart software based on an open-source algorithm called “Pyphysio” [87], developed in “Python” by the Bruno Kessler Foundation (Italy), University of Trento (Italy), and Nanyang University of Technology (Singapore). Pyphysio enhances the quality of ECG analysis to quantify physiological effects using different types of algorithms for processing ECG signals, evaluation of signal quality, temporal alignment of multiple signals, and extraction of indicators of sequential segments and post-processing analysis. The physiological signal processing is based on the Pyphysio algorithm that spans three subclasses (filter, estimator, and indicator). Each subclass is dedicated to a phase in a typical signal processing for psychophysiology: 1- filter (a subclass of algorithms for preprocessing for removing noise and improving signal quality); 2- estimator (a subclass of algorithms that collects all the algorithms that aim to extract the critical information from the input signal); 3- indicator (a subclass of mathematical functions that compute metrics or scalar values from the input signal like the energy at a specific frequency band).

### 2.4. Procedure

Children and adolescents with ASD were assessed individually in 2019 at a Spanish psychological center through the RIAS intelligence test, the Bears Family Projective Test, and the SENA questionnaire (self-report form).

The behavioral and functional data were obtained from families and teachers through the SENA questionnaire (family and school forms) in previous sessions.

In the first 45-min individual session with each participant, the first part of the RIAS intelligence test and the Bears Family Projective Test were administered. In a second individual session, the second part of the RIAS intelligence test and the SENA questionnaire (self-report form) were administered.

For the stories told by the participants using the Bears Family Projective Test, two trained researchers, blinded to the research hypotheses, coded the video-recorded story sessions using the Bears Family Test Manual [83]. The reliability was evaluated in 50% of the sample using Cohen’s kappa index, which was statistically acceptable (kappa = 0.89).

The discursive dialogic online therapy sessions were offered during SARS-CoV-2 pandemic lockdown in Spain (March–June 2020), collecting data from two different sources: 1—he video recording through OBS software (obsproject.com) of each online psychotherapy session via Skype Video Call (skype.com); and 2—the therapist’s heart rate activity recording through the Comftech CozyBaby ECG sensor (comftech.com), the signal acquisition PhysioREC software (physiolitix.fbk.eu), a smartphone Xiaomi RedmiNote 3, and the ECG signal processing system Pyphysio [87].

The protocol used during the sessions to capture both recordings was: 1- the smartphone was connected to the ECG sensor using Bluetooth; 2- the application PhysioREC was opened to upload ECG data to a web platform; 3- the electrodes were moistened to improve sensibility; 4- the sensor’s band was set on the therapist’s chest between the manubrium and the body sternum; 5- the recording of the camera and ECG sensor were started.

For the analysis of the turn-taking exchange between the therapist and each patient during the online therapy sessions, two independent researchers, blinded to the study hypothesis, coded the video recorded sessions using the turn-taking sequential observation system, considering minimum sequences of one second. The reliability was evaluated in 65% of the sample using Cohen’s kappa index [89], which was statistically acceptable (kappa = 0.87).

For the analysis of the turn-taking exchange between the therapist and each patient during the online therapy sessions, two independent researchers, blinded to the study hypothesis, coded the video-recorded sessions using the turn-taking sequential observation system, considering minimum sequences of one second. The reliability was evaluated in 65% of the sample using Cohen’s kappa index, which was statistically acceptable (kappa = 0.87).

For the analysis of the therapist’s heart rate variability (HRV), proceeding from the ECG sensor during each session, a researcher, blinded to the study hypothesis, analyzed the ECG signal, and extracted some key features (RR Interval, Poincaré SD1, and SD2, and Beats per second) to detect high and low heart HRV. The RR interval, also known as the interbeat interval (IBI), is the time between one beat and the next using the ECG signal’s R peak as a reference [90]. The Poincaré analysis is a nonlinear method to assess the HRV getting through the Poincaré plot, a diagram with SD1 (the length of the ECG wave) on the vertical axis, and the SD2 (the longitude of the ECG wave) on the horizontal axis. Each point in the Poincaré plot is the R-R interval, a function of the previous R-R interval and the successive R-R interval.

We used this measurement’s mean and standard deviation to calculate the Poincaré SD1 and SD2 that associate HRV to stress levels. HRV reflects the variation of the beat-to-beat (RR) intervals and is an indicator of the autonomic nervous system activity [91]. SD1 represents the fast RR variability in the HRV data, while SD2 describes the long-term variability (SD1 and SD2 are also known as the coefficients of the Poincaré plot), and SD1/SD2 is the ratio of short interval variation to the long interval variation [92]. To perform these ECG signal analyses, we used the PyPhysio library [87].

Previously to calculate the features shown above, the signal’s quality was checked morphologically, looking for noise or fragments of the corrupted recording. From a total of the 33 recordings, seven were discarded due to the presence of noise or corrupted fragments. The final number of samples was 26, with enough quality to be analyzed. We obtained all the features using a time window of 24 s. Each second of the ECG signal it was ranged from current time point ±12 s.

To study the therapist’s different stress levels, we clustered the different sessions in two groups (low and high stress level), according to the SD1 and SD2 parameters, related inversely to the sympathetic activity. Some authors also use the activity stress score SS based on SD2 (SS = 1000 × 1/SD2; Dishman et al. 2000) [93] and, for this purpose, we used the k-means method technique with a k = 2. SD2 is a measure of a low variation in heart rate (low HRV) that reflects a low sympathetic activation and a low level of stress [94]. This measurement consists of analyzing the time variations between consecutive beat and beat [95,96]. In a resting state, the influence of the parasympathetic nervous system predominates on the heart (low sympathetic activity–low HRV–low stress). In contrast, in a situation of stress, anxiety, or physical exercise, the sympathetic nervous system’s activity predominates (high sympathetic activity–high HRV–high stress [93]).

Consequently, a low HRV reflects a low sympathetic activity (low-stress), while a high HRV reflects a high sympathetic activity (high-stress). Table 2 shows the results of this clustering process.

### 2.5. Data Analysis

To analyze the relationships between the communicative intention during the turn-taking of children/teenagers with ASD and therapist (Hypotheses 1 and 2), we run Pearson correlation (95% and 99% confidence levels with a two-tailed test of significance).

To explain the differences between the heart rate variability (HRV) of the therapist and patients’ events (Hypotheses 3). First, we performed Shapiro–Wilk normality tests on the two categories of study variables (high and low HRV, sample size smaller than 50). Then, we performed a t-test to determine the equality of means on those variables that meet the normality assumption in both categories (high and low HRV) and a nonparametric test (Mann–Whitney) on those variables that do not meet the normality criterion in both categories.

Additionally, to study the impact—both on the patient and the therapist—of the different events during the session on the SD2 parameter, we used the nonparametric test hypothesis for k independent samples (Kruskal–Wallis test) grouped by therapist and patient events.

## 3. Results

### 3.1. Changes in Turn-Taking per Second, Participants’ Intelligence, Age, Narrative, and Social Skills

Concerning Hypothesis 1, according to which the communicative intention during the turn-taking in the therapeutic session can be related to participants’ intelligence, age, narrative, and social skills, the results confirm it (Figure 1 and Figure 2). below describe the evolution of the variables related to this hypothesis during the twenty-six online psychotherapy sessions: frequency of turn-taking per second (RATIO), general IQ, Verbal IQ, chronological age, equivalent age, Bears Family cohesion and structure indexes. We obtained the frequency of turn-taking per second (RATIO) by dividing the total number of turn-taking exchanges during the session (sequential observation system [85,86]) by the entire duration of the session expressed in seconds. The general IQ and the Verbal IQ refer to the total IQ and the verbal IQ scores, respectively, obtained by the participant in the RIAS intelligence test [79]. The chronological and equivalent age refers to actual age and the equivalent age based on IQ assessment. The Bears Family narrative cohesion index regards the participant’s ability to tell a story with a perspective, around a theme or a problematic element, with the Bears Family Projective Test [70,81,82,83,84] on a Likert scale (0–11). The Bears Family structure index belongs to the participant’s capability to tell a story, with the introductory phrase, the protagonist, the time and physical setting, the short-term and long-term conclusion, with the Bears Family Projective Test [70,81,82,83,84], on a Likert scale (0–6) (Figure 1 and Figure 2).

Descriptive statistics about session duration, the number of turn-taking events, and the turn-taking frequency per second are shown in Table 3.

The results indicate that a higher IQ matches more turn-taking events between patient and therapist (r = 0.53, *p* = 0.05), considering the frequency of changes in turn-taking per second (Figure 1). Moreover, the age results associated with higher cohesion (chronological age: r = 0.44, *p* = 0.05; equivalent age: r = 0.52, *p* = 0.01) and structure (chronological age: r = 0.50, *p* = 0.01; equivalent age: r = 0.38, *p* = 0.07) of the patient’s story with the Bears Family Projective Test (Figure 2).

At a younger age, the patient accepts (patient accepts—NA) the therapist’s initial conversation topic proposal for a longer time (r = −0.50, *p* = 0.05); meanwhile older patients expands (patient Expands—NAm) more on the therapist’s conversation during the session (r = 0.40, *p* = 0.05) (Table 4).

Concerning narrative competence, patients who told stories with a more significant number of propositions at the Bears Family Projective Test, spent more time expanding (patient expands—NAm) the therapist’s conversation during the session (r = 0.46, *p* = 0.05).

Patients who reached a greater story cohesion index in the Bears Family Projective Test spent less time, during the session, not sharing and continuing with their activity (patient does not share and continues with their topic—NNcC; r = −0.51, *p* = 0.01).

At a low perception of self-esteem (self-report, r = −0.57, *p* = 0.01), less integration and social competence perceived by the patient (self-report, r = −0.55, *p* = 0.05) corresponded a mayor frequency of times with the patient not accepting the initial proposal of the therapist (patient does not accept—NNA).

A lower perception of personal resources at home corresponded with longer time, during the session, that the patient did not share and propose a topic to the therapist (patient does not share and proposes—NNcP; parental report, r = −0.47, *p* = 0.05).

### 3.2. Participants’ Difficulties and Turn-Taking with the Therapist

Concerning Hypothesis 2, according to which the turn-taking between the therapist and the young patients with ASD can be related to the participant’s emotional and behavioral difficulties, according to self-report, family, and school, the results confirm it.

On the one hand, the therapist takes more time to start the online therapy session (therapist proposes—T*) with patients who perceive and show at school more behavioral problems (self-report, r = 0.48, *p* = 0.05; teacher’s report r = 0.59, *p* = 0.05) and who perceive and show at home fewer personal resources (self-report, r = −0.65, *p* = 0.01; parental report, r = −0.51, *p* = 0.01).

The therapist sustains (therapist holds—TS) for a longer time the patient’s conversation topic with patients who perceive more aggressive behaviors (self-report, r = 0.49, *p* = 0.05), show more global problems (parental report; r = 0.43, *p* = 0.05) and emotional regulation difficulties at home (parental report, r = 0.50, *p* = 0.05). Moreover, the therapist stays for a longer time co-oriented (co-oriented therapist-TMC) with patients who perceive and show at home more aggressive behaviors (self-report, r = 0.70, *p* = 0.01; parental report, r = 0.43, *p* = 0.05).

The therapist directs for a longer time attention (therapist directs attention—TDAt) when interacting with patients who perceive more behavioral problems (self-report, r = 0.43, *p* = 0.05), and show, at school, more global problems (teacher’s report r = 0.84, *p* = 0.01), emotional problems (teacher’s report, r = 0.61, *p* = 0.05), behavioral problems (teacher’s report, r = 0.79, *p* = 0.01), difficulties in executive functions (teacher’s report, r = 0.90, *p* = 0.01), depressive features (teacher’s report, r = 0.56, *p* = 0.05), hyperactivity (teacher’s report, r = 0.90, *p* = 0.01), anger (teacher’s report, r = 0.67, *p* = 0.01), aggressiveness (teacher’s report, r = 0.59, *p* = 0.05), challenging behavior (teacher’s report, r = 0.91, *p* = 0.01), unusual behaviors (teacher’s report, r = 0.72, *p* = 0.01), emotional regulation difficulties (teacher’s report, r = 0.75, *p* = 0.01), and rigidity (teacher’s report, r = 0.78, *p* = 0.01).

The therapist interrupts the conversation for a longer time and proposes a new conversation topic (therapist interrupts—TI) when interacts with patients who perceive more antisocial behaviors (self-report, r = 0.70, *p* = 0.05), less emotional regulation at home (parental report, r = −0.40, *p* = 0.05) and, at school, more global problems (teacher’s report, r = 0.80, *p* = 0.01), behavioral problems (teacher’s report, r = 0.82, *p* = 0.01), executive dysfunctions (teacher’s report, r = 0.71, *p* = 0.01), depressive features (teacher’s report, r = 0.61, *p* = 0.05), attention difficulties (teacher’s report, r = 0.59, *p* = 0.05), hyperactivity (teacher’s report, r = 0.68, *p* = 0.01), anger (teacher’s report, r = 0.71, *p* = 0.01), aggressiveness (teacher’s report, r = 0.80, *p* = 0.01), behavior challenging (teacher’s report, r = 0.81, *p* = 0.05), unusual behaviors (teacher’s report, r = 0.68, *p* = 0.01), and emotional regulation problems (teacher’s report, r = 0.74, *p* = 0.01).

Finally, the therapist takes longer to finish the online session (therapist ends—TT) with patients who perceive and show at home and school more behavioral problems (self-report, r = 0.48, *p* = 0.05; parental report, r = 0.58, *p* = 0.01; teacher’s report, r = 0.60, *p* = 0.05), and anger (self-report, r = 0.49, *p* = 0.05; parental report, r = 0.50, *p* = 0.05; teacher’s report, r = 0.61, *p* = 0.05).

On the other hand, the patient proposes a topic to the therapist (child proposes—N*) for a longer time when they perceive more aggressive behaviors (self-report, r = 0.49, p = 0.05), and expands (child Expands—NAm) the conversation longer when they perceive more difficulties with peers (self-report, r = 0.48, *p* = 0.05), and he perceive and show at home fewer emotional problems (self-report, r = −0.60, *p* = 0.01; parental report, r = −0.55, *p* = 0.01). Despite it, patients who proposes a topic to the therapist for a longer time also perceive more antisocial behaviors (self-report, r = 0.61, *p* = 0.05), and show at home more aggressive behaviors (self-report, r = 0.71, *p* = 0.01; parental report, r = 0.43, *p* = 0.05).

The patients who did not share with therapist and continued with their conversation or action for longer (child does not share and continues with his activity—NNcC) were those who used to show at school more global problems (r = 0.56, p = 0.05), emotional problems (r = 0.58, *p* = 0.05), executive dysfunction (r = 0.64, *p* = 0.01), depressive features (r = 0.53, *p* = 0.05), hyperactivity (r = 0.66, *p* = 0.01), challenging behavior (r = 0.61, *p* = 0.05), stiffness (r = 0.76, *p* = 0.01), isolation (r = 0.51, *p* = 0.05), and fewer personal resources (r = −0.53, *p* = 0.05).

### 3.3. Therapist’s Heart Rate Variability (HRV) and Patient’s Turn-Taking Synchrony

Concerning Hypothesis 3, according to which during the psychotherapeutic session, the therapist’s heart rate variability (HRV) can be related to the patient’s turn-taking synchrony, with more significant stress on the therapist throughout shared and broadened activities, the results confirm it.

Considering the global average of the SD2 ECG parameter (HRV) of the therapist (Table 5) during the 10 low-stress (group 1) and the 16 high-stress sessions (group 2) the unique difference detectable is the frequency of attempts to end the session (therapist ends -TT; mean group 1 = 1.2; SD group 1 =0.6; mean group 2 = 0.56; SD group 2 =0.51; chi2 7.09; gl 2; p 0.05). It indicates that considering the therapist’s SD2 HRV mean of the whole session, the lower was the therapist’s frequency of attempts to terminate the session the higher was his sympathetic activity (high-stress level).

Hence, we analyzed the relationship between the SD2 parameter, and the different actions taken, by therapist and patient, during the turn-taking interactions in the whole session, searching for significant differences. Table 6 and Table 7 show the obtained descriptive statistics.

Once it is verified that the data do not follow a normal distribution (Table 8), we performed the nonparametric test hypothesis for k independent samples (Kruskal Wallis Test) for each table independently (considering the therapist’s SD2 grouped by therapist’s and patient’s interactive behavior, Table 6 and Table 7) obtaining significant differences that confirms that the taken actions considered are related with the therapist SD2 ECG parameter (Table 9).

The SD2 values are then sorted for a better understanding and interpretation of their meaning. We grouped the therapist’s SD2 parameter by the therapist’s turn-taking interactive behavior. Considering that a lower SD2 corresponds to a higher therapist HRV (high sympathetic activity, high-stress level), we listed in Table 6 and Table 7 the therapist’ SD2 in decreasing order (from high-stress to low-stress), according to the therapist’s (Table 6) and patient’s (Table 7) turn-taking action. So, we registered higher therapist’s HRV (high sympathetic activity, high-stress level, Table 6) in the following sequence: 1st TMC (Co-oriented therapist), 2nd TA (therapist expands), 3rd TDAt (therapist directs attention), 4th TDAc (therapist directs action), 5th TI (therapist interrupts), 6th T* (therapist proposes).

Similarly, we grouped the therapist’s SD2 parameter by the patient’s turn-taking interactive behavior (Table 7). So, we registered higher therapist’s HRV (high sympathetic activity, high-stress level, Table 7) in the following sequence: 1st Nam (patient expands), 2nd NC (patient shares), 3rd NNcC (patient does not share and continues with their topic), 4th NNcP (patient does not share and proposes), 5th NA (patient accepts), 6th NNA (patient does not accept).

These results indicate that there is a more significant stress on the therapist both when they shares (TMC) and expand (TA) the conversation with the patient both when the patient expands (Nam) and shares (NC) the conversation with the therapist.

## 4. Discussion

### 4.1. Changes in Turn-Taking per Second, Participants’ Intelligence, Age, Narrative, and Social Skills

The study aimed to analyze the therapist’s physiological reactions and conversational turn-taking during online psychotherapy with a group of children and adolescents with ASD at verbal level 1. We considered the patient’s intellectual and narrative skills, emotional and behavioral difficulties from a multi-informant perspective (patient, family, and school).

Hypothesis 1 hypothesized that the communicative intention during the turn-taking of children/teenagers with ASD and therapist could be related to participants’ intelligence, age, narrative, and social skills.

According to Hypothesis 1, the results confirmed that the communicative intention during the turn-taking between patient and therapist was related to the patient’s age and socio-cognitive skills. The older the patient was, and the more he told stories with more proposition and adaptive behaviors in the narrative baseline using the Bears Family Projective, the more conversation he expanded during the online session with the therapist. Moreover, the patients who self-reported several personal resources and greater self-esteem, expanded more, and shared the conversation during the online session, producing the therapist’s most physiological activation (HRV) in the therapist.

At a lower equivalent and chronological age, the patient accepted the initial therapist proposal for longer at the beginning of the session; meanwhile, at a lower self-esteem perception, integration, and social competence, the patient did not accept the initial therapist proposal as long.

Generally, according to self-report, acceptance in turn-taking during all the psychotherapeutic sessions was longer with patients who perceived fewer emotional and contextual problems, depressive features, and anxious symptoms; according to the family perception, with fewer emotional problems, difficulties in executive function, depressive, anxiety, rigidity, isolation, but more significant behavioral and challenging problems; and, according to the teacher perception, with more challenging behavior, learning problems, and difficulties in emotional regulation.

In highly charged cognitive and emotional circumstances, the person’s capacity to manage and regulate their behavior is minimized [97,98,99,100]. Therefore, a difficulty in emotional processing may imply an overproduction of unresolved narrative representations and maladaptive behaviors, which are not balanced by resolved and positive elements [81,84]. Recent studies emphasize problem-solving competence in developing executive functions, and emotional and behavioral self-regulation [88,98,101,102]. From this perspective, conversational turn-taking with a specialized professional can underline the relationship between patients’ communicative intention skills, emotional regulation, self-esteem, and behavioral competencies throughout the narrative and shared conversation during the psychotherapeutic session.

These findings indicate that the mental health specialist must promote balanced positive and negative representations in the conversation about oneself and the patient, past, present, and future experiences, alternating and balancing own and other representations, a primary objective in dialogic psychotherapy, especially with children and adolescents with ASD at the verbal level [35,103,104,105].

### 4.2. Participants’ Difficulties and Turn-Taking with the Therapist

Hypothesis 2 hypothesized that the turn-taking between the therapist and the young patients with ASD could be related to the participant’s emotional and behavioral difficulties, according to self-report, family, and school.

For Hypothesis 2, the results confirmed that conversational turn-taking between therapist and patient was related to the patient’s emotional and behavioral difficulties. Indeed, the therapist spent more time starting online therapy sessions, sustaining, co-orienting, directing attention, interrupting the conversation, proposing a new topic, and finishing the session with patients with more behavioral problems and fewer personal resources. On the other hand, patients with more behavioral problems spent more time proposing a topic to the therapist, expanding, and not sharing his conversational topic or action.

These findings indicate that it is more challenging to open and alternate therapist’s own and patient’s representations when the patient shows more behavioral difficulties. The specialist’s objective must be to share an argument in conversation to balance positive and negative representations, facilitating both metacognitive and executive skills. In this case, a more active role for the therapist is necessary, interrupting and directing the conversation to promote coherence between the patient’s dialogic representations and his owns, promoting a shared communicative exchange that actively involves both interlocutors [88,106,107,108].

### 4.3. Therapist’s Heart Rate Variability (HRV) and Patient’s Turn-Taking Synchrony

Hypothesis 3 hypothesized that, during the psychotherapeutic session, the therapist’s heart rate variability (HRV) could be related to the patient’s turn-taking synchrony, with more significant stress on the therapist throughout shared and broadened activities.

Concerning Hypothesis 3, the results confirmed that the therapist’s heart rate variability (HRV) was related to turn-taking during the interactions with the patient, with more significant stress on the therapist both when he shared and expanded the conversation with the patient both when the patient expanded and shared the conversation with the therapist.

This result underlines that the therapist must adapt to the patient’s peculiarities, looking for an active strategy to support and expand the patient’s conversation and topic proposals in a balanced and coherent way. Like a mirror of the patient’s effort to enter and remain in the shared dialogic contact, the therapist experiences a heart rate stress, probable reflex of emotional resonance that he must recognize, regulate, and reorganize to return to the therapeutic relationship in a modulated and understandable way, operating an emotional attunement with the patient [2,17]. During the online session, the therapist’s physiological activation is more significant when the situation requires higher synchrony with the patient to favor emotional regulation and balanced representations on a shared topic. In this way, synchrony requires a dynamic and reciprocal adaptation of interlocutors’ behaviors, promoting an emotional and conversational exchange between interactive partners [109].

Synchrony is a social signal, and synchrony improvement can lead to cognitive and behavioral improvement in young patients with ASD [7,109]. During the interaction process, emotional activation implies several modifications at the somatic, neuroendocrine, vegetative, perceptual, and cognitive–representational levels. If cognitive processes are challenging to investigate, the human organism’s autonomous vegetative activity can be observed directly by measuring peripheral psychophysiological activation such as heart rate variability [2,17].

The psychophysiological response is triggered in interpersonal interactions, where cognitions, emotions, and empathy feed physiological responses of approach or flight [93,110]. Neuroaffective, sensorial, attachment and personality traits determine, to a large extent, the modality through which an individual meets another in a specific intersubjective exchange [1,2,9,17,111].

So emotional fluctuations can be shared and linked in a relational scaffolding through emotional and cognitive-relational exchanges during specific relationship experiences such as dialogic psychotherapy. Psychotherapy, presential or online, represents a particular intersubjective interaction context. The high level of confidentiality and affective involvement allows synchronous interlocutors to visit past/present experiences and anticipating future situations, looking to balance positive and negative representations, being an active emotionally involved part in a turn-taking interaction process [2,17].

## 5. Conclusions

Several therapeutic approaches with ASD aim to develop the patient’s functional speech [112]. Some of the techniques used during online dialogic psychotherapy can increase motivation within an ecological environment for the patient, structuring conversation throughout a psychotherapeutic scaffolding, using several stimuli variations, reinforcements, and facilitating the patient’s verbal communication attempts [112]. In this online setting, dialogic therapy with ASD can increase spontaneity, the use of language, and social and communicative motivation [113].

This study contributes to the lack of knowledge about the turn-taking and communicative processes between therapists and young patients with ASD. However, synchrony is an essential aspect for the prognosis of the child and adolescent with ASD [103,104], feeding the study about clinical and psychotherapeutic practice [107].

Thus, it is essential to promote the child’s communication because of its functional nature. Communication that occurs naturally always has a goal and is therefore closely related to the motivation of the person who wants to communicate. Children and adolescents with ASD show specific difficulties interacting with other people because they often experience these interactions as frustrating and not always successful. Therefore, the therapist or person who fosters a successful interaction must be sensitive and not hostile with children and adolescents with ASD [114] sharing interests and providing positive and balanced relational experiences, sharing a real interest towards the patient’s discourse, motivating, and enriching the psychotherapeutic process [115].

## Figures and Tables

**Figure 1 brainsci-11-00586-f001:**
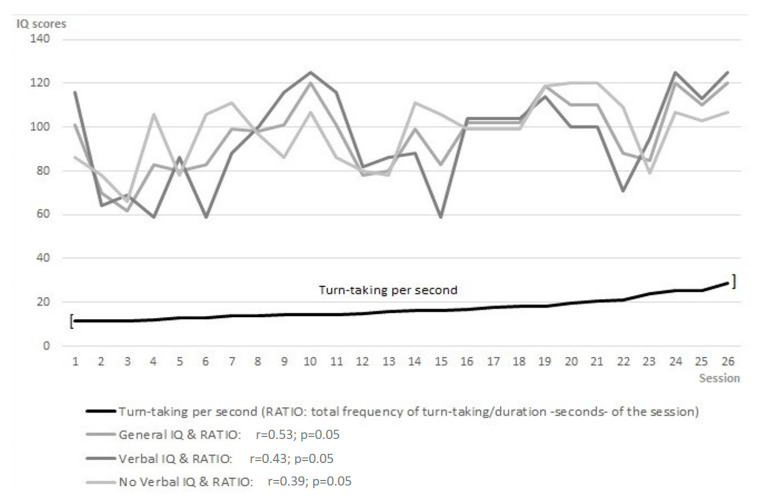
Frequency of changes in turn-taking per second and participant’s IQ.

**Figure 2 brainsci-11-00586-f002:**
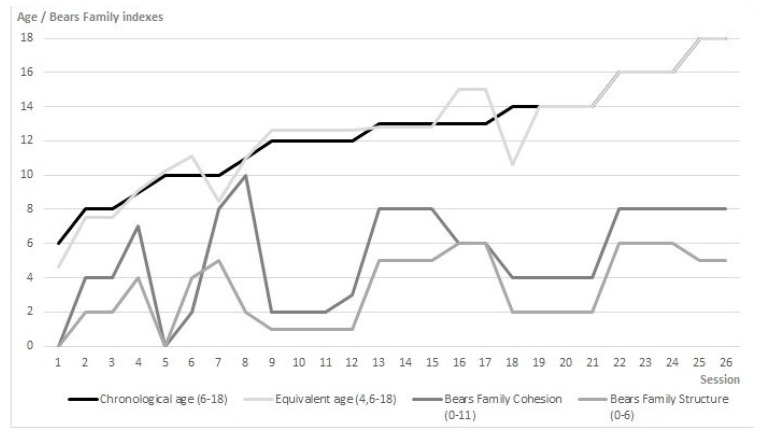
Participant’s age, story cohesion, and structure achieved in the Bears Family Projective Test.

**Table 1 brainsci-11-00586-t001:** Characteristics of the participants, n = 16, Males.

	Min	Max	Average	SD	Skewness	Kurtosis
Chronological age—years	6	18	12.0	3.00	−0.12	−0.13
Equivalent age (IQ-RIAS)—years	4.6	18	12.50	3.29	−0.45	0.08
General IQ (IG-RIAS) ^①^	62	120	96.38	16.01	−0.25	−0.64
Verbal IQ (IV-RIAS) ^①^	59	125	94.92	21.50	−0.34	−1.01
Non-verbal IQ (INV-RIAS) ^①^	66	120	97.81	15.06	−0.40	−0.86
Verbal memory IQ (IM-RIAS) ^①^	60	120	97.23	14.48	−0.49	0.82
ADOS/ ADOS-2 module ^②^	2	4				
Total ADOS/ ADOS-2 score ^②^	7	12	10	1.37	0.43	−0.36
Number of propositions (Bears Family story)	0	75	23.08	23.58	1.37	0.42
Number of episodes (Bears Family story)	0	40	11.00	10.53	1.86	3.36
Cohesion index (Bears Family story) ^③^	0	10	5.00	2.87	−0.10	−1.11
Structure index (Bears Family story) ^④^	0	6	3.17	2.12	0.12	−1.56
Global problem index (SENA-self report) ^⑤^	39	81	55.48	10.80	0.39	0.09
Emotional problems (SENA-self report) ^⑤^	40	76	55.35	12.09	0.33	−1.55
Behavioral problems (SENA-self report) ^⑤^	41	88	55.35	15.03	1.02	−0.03
Executive functions problems (SENA-self report) ^⑤^	39	79	53.70	10.99	0.45	−0.15
Personal resources (SENA-self report) ^⑤^	15	54	38.65	9.75	−0.58	0.58
Self-esteem (SENA-self report) ^⑤^	21	62	44.70	9.67	−0.29	0.40
Global problem index (SENA-family report) ^⑤^	44	82	63.12	11.73	0.33	−1.16
Emotional problems (SENA-family report) ^⑤^	36	84	61.88	15.85	−0.33	−1.11
Behavioral problems (SENA-family report) ^⑤^	40	90	57.80	13.97	0.85	0.26
Executive functions problems (SENA-family report) ^⑤^	53	83	65.16	10.37	0.58	−1.12
Personal resources (SENA-family report) ^⑤^	21	50	34.64	8.32	0.02	−0.44
Global problem index (SENA-teacher report) ^⑤^	44	83	56.56	12.45	1.32	0.87
Emotional problems (SENA-teacher report) ^⑤^	46	64	53.63	6.37	0.27	−1.44
Behavioral problems (SENA-teacher report) ^⑤^	44	88	54.69	14.30	1.75	2.36
Executive functions (SENA-teacher report) ^⑤^	50	84	59.31	12.43	1.25	0.14
Personal resources (SENA-teacher report) ^⑤^	13	46	32.50	10.48	−0.11	−1.21

^①^ IQ scores: mean of 100 and standard deviation of 15. ^②^ ADOS/ADOS-2 modules and Cutoff scores for ASD: Module 2 (Phrase Speech), Autism (Total Score ≥ 10), Autism Spectrum (Total Score ≥ 7); Module 3 (Fluent Speech, Child/Adolescent), Autism (Total Score ≥ 9), Autism Spectrum (Total Score ≥ 7); Module 4 (Fluent Speech, Adolescent/Adult), Autism (Total Score ≥ 10), Autism Spectrum (Total Score ≥ 7). ^③^ Bears Family Cohesion Index: scalar score 1–11. ^④^ Bears Family Structure Index: scalar score 1–6. ^⑤^ T-scores: mean of 50 and standard deviation of 10.

**Table 2 brainsci-11-00586-t002:** Clustered patients according to normalized SD2 parameter of the therapist’s ECG [91].

Group	SD2 Mean/SD
Group 1 (Low Sympathetic Activity –Low Stress Level)	334,754/58,964
Group 2 (High Sympathetic Activity –High Stress Level)	92,518/38,868

**Table 3 brainsci-11-00586-t003:** Therapy session duration and frequency of turn-taking per second (n = 16 patients, n = 26 therapy sessions).

Online Therapy Session	Min	Max	Average	SD	Skewness	Kurtosis
Session duration (seconds)	773	3000	1851.92	667.01	0.01	−1.17
Number of turn-taking events (frequency)	38	202	117.69	51.06	−0.15	−1.22
Ratio—session duration (seconds)/total number of turn-taking events (frequency)	11	29	16.96	4.75	0.94	0.30

**Table 4 brainsci-11-00586-t004:** Patient’s characteristics and turn-taking events, (n = 16 patients, n = 26 therapy sessions).

Patient’s Characteristics	Child Accepts (NA)	Child Expands (NAm)	Patient Does Not Share (NNcC)	Patient Does Not Accept (NNA)	Patient Does Not Share and Proposes (NNcP)
Chronological age	r = −0.50 *p* = 0.05	r = 0.40 *p* = 0.05			
Number of propositions (Bears Family story)		r = 0.46 *p* = 0.05			
Cohesion index (Bears Family story)			r = −0.51 *p* = 0.01		
Self-esteem (SENA-self report)				r = −0.57 *p* = 0.01	
Personal resources (SENA-parental report)					r = −0.47 *p* = 0.05

**Table 5 brainsci-11-00586-t005:** Therapist’s frequencies of attempts to end the session clustered according to normalized SD2 parameter of the therapist’s ECG HRV.

Therapist’s Behavior	Group	Mean	SD
Therapist ends (TT) - behavioral frequency -	Group 1 (Low Sympathetic Activity—Low Stress)	1.2	0.63
	Group 2 (High Sympathetic Activity—High Stress)	0.56	0.51

**Table 6 brainsci-11-00586-t006:** The Therapist’s SD2 statistics grouped considering the therapist’s interactive behavior. A lower SD2 corresponds to a higher therapist’s HRV (high sympathetic activity, high-stress level).

Behavior	Mean	Median	Variance	Std. Dev.	Min.	Max.	Range	Skewness	Kurtosis
Co-oriented therapist	327.32	318.31	23,587.96	153.58	21.34	813.49	792.16	0.42	−0.05
Therapist expands (TA)	332.71	332.81	20,231.79	142.24	21.34	891.48	870.15	0.06	−0.05
Therapist directs attention (TDAt)	411.73	462.02	18,996.30	137.83	135.46	625.71	490.24	−0.56	−0.99
Therapist directs action (TDAc)	429.52	451.02	18,178.90	134.83	37.02	719.27	682.25	−1.34	2.11
Therapist interrupts (TI)	458.45	518.71	15,643.51	125.07	248.08	630.02	381.94	−0.92	−0.59
Therapist proposes (T*)	537.27	580.35	15,435.04	124.24	367.92	686.61	318.69	−0.51	−1.57

**Table 7 brainsci-11-00586-t007:** The therapist’s SD2 statistics grouped considering the patient’s interactive behavior. A lower SD2 corresponds to a higher therapist HRV (high sympathetic activity, high-stress level).

Behavior	Mean	Median	Variance	Std. Dev.	Min.	Max.	Range	Skewness	Kurtosis
Patient expands (NAm)	329.97	321.05	23,810.13	154.31	21.34	813.49	792.16	0.39	−0.12
Patient shares (NC)	334.62	335.04	20,277.21	142.40	21.34	891.48	870.15	0.07	−0.01
Patient does not share (NNcC)	368.53	407.46	25,309.95	159.09	42.17	719.27	677.10	−0.48	−0.88
Patient does not share and proposes (NNcP)	383.77	394.58	13,807.32	117.50	198.01	557.53	359.53	−0.29	−1.22
Patient accepts (NA)	461.70	440.47	6395.40	79.97	367.92	605.74	237.82	0.61	−0.98
Patient does not accept (NNA)	618.49	614.61	1983.16	44.53	563.59	686.61	123.02	0.20	−1.39

**Table 8 brainsci-11-00586-t008:** Kruskal–Wallis test SD2 therapist’s HRV grouping for therapist and patient turn-taking interactive behaviors.

			Kolmogorov-Smirnov			Shapiro-Wilk		
			Statistic	df	Sig.	Statistic	df	Sig.
Therapist	SD2	M*	0.251	15	0.012	0.823	15	0.007
		MA	0.023	1808	0.025	0.994	1808	0.000
		MC	0.047	1300	0.000	0.983	1300	0.000
		MDAc	0.189	86	0.000	0.868	86	0.000
		MDAt	0.176	113	0.000	0.904	113	0.000
		MI	0.229	22	0.004	0.807	22	0.001
Patient	SD2	NA	0.185	22	0.048	0.886	22	0.016
		NAm	0.045	1318	0.000	0.984	1318	0.000
		NC	0.023	1729	0.039	0.994	1729	0.000
		NNA	0.133	10	0.200	0.935	10	0.501
		NNcC	0.126	185	0.000	0.936	185	0.000
		NNcP	0.116	80	0.009	0.913	80	0.000

**Table 9 brainsci-11-00586-t009:** Kruskal Wallis Test SD2 therapist’s HRV grouping for therapist and patient turn-taking interactive behaviors.

	Therapist SD2 (HRV) Grouped by Therapist’s Turn-Taking Interactive Behavior. (TMC, TA, TDAt, TDAc, TI, T*)	Therapist SD2 (HRV) Grouped by Patient’s Turn-Taking Interactive Behavior. (NAm, NC, NNcC, NNcP, NA, NNA)
Chi-Square Statistical	124.710	75.292
Asymp. Sig.	**0.001**	**0.001**

## Data Availability

The dataset is released under the Creative Common 4.0 (CC-BY) License at https://doi.org/10.6084/m9.figshare.14185748 (accessed on 29 April 2021).

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
