# Peer review of "Physiological Reactions in the Therapist and Turn-Taking during Online Psychotherapy with Children and Adolescents with Autism Spectrum Disorder"

_brainsci, 2021, doi:10.3390/brainsci11050586_

Round 1

Reviewer 1 Report

This is a very interesting study and well thought out.  I noticed a few errors in English grammar and would recommend a minor edit of English.

Reviewer 2 Report

The paper is interesting because to analyze the relationship between the social skills of children and adolescents with Autism Spectrum Disorder, the variability of therapist's heart rate and the conversational turn-taking during online psychotherapy. The results show that the patients' communicative intention was related to their narrative, intellectual and social competencies and that the turn-taking between the therapist and the participant was associated with the patient's emotional and behavioral difficulties. So, the therapist's heart rate was related to their synchrony with the patients.  The study is well conducted, and the interpretation of the results sounds. Despite that, I have a few major and minor concerns that need to be addressed before accepting the manuscript for publication. In particular, the explication of the rationale and the description of results are improvable.

  1. Sample

A critical point is the verbal level of children, because it is very important for adequate conducting the psychotherapy intervention, that is based on conversational, linguistic, metalinguistic and socio-communication abilities. In the Participants section, it is highlighted that children with ASD in this study have a verbal level 1. ASD children with verbal level 1 are able to speak in full sentences and communicate, but has trouble engaging in back-and-forth conversation with others. Typically, children with this level of language abilities require few supports and show a “high-functioning" in cognitive performances. This point is in conflict with data in Table 1, because it is reported a very wide range both as regards the general cognitive level and as regards the verbal level. The sample shows also the presence of children with general and verbal IQ in the context of mild intellectual disability. Furthermore, the chronological age of children also seems to be a weak point, given that the training was also undertaken with very young children with ASD (6 years), who may have had a lot of difficulties in accessing to some types of cognitive and emotional reasonings (hard also for children with typical development at this age). Finally, the ADOS scores could also be an important index because it makes better understand the severity of autistic symptoms, but those scores are not reported.

  1. Assessment

The authors used the Reynolds RIAS intelligence test, rather than other types of cognitive tests (for example the Weschler Scales). Why? The use of this tool with ASD children and adolescents is not so frequent in the scientific literature. Furthermore, the evaluation of adaptive functioning (for example with Vineland Scales) is not reported, but it is a very important information to understand the correct modality of interaction with this population.

  1. Results

Tables are not very clear, and it is very laborious to detect and understand the statistically significant/non-significant data. Furthermore, the use of some graphic representation of results could help their comprehension.

  1. Discussion

It could be interesting to examine in depth the possible neuropsychological insights for the treatment of this population of patients. It could be useful to divide the discussion (and also the results) into sections to better follow the authors' reasoning.

Round 2

Reviewer 2 Report

ok

Author Response

Thank you for your constructive reviews.